# Risk Perception of the SARS-CoV-2 Pandemic: Influencing Factors and Implications for Environmental Health Crises

**DOI:** 10.3390/ijerph20043363

**Published:** 2023-02-14

**Authors:** Timothy Mc Call, Susanne Lopez Lumbi, Michel Rinderhagen, Meike Heming, Claudia Hornberg, Michaela Liebig-Gonglach

**Affiliations:** 1Medical School OWL, Bielefeld University, 33615 Bielefeld, Germany; 2Institute for Occupational, Social and Environmental Medicine, Centre for Health and Society, Faculty of Medicine, Heinrich Heine University of Düsseldorf, 40225 Düsseldorf, Germany

**Keywords:** risk perception, environmental health, climate change, SARS-CoV-2

## Abstract

Background: The SARS-CoV-2 pandemic and climate change are two simultaneously occurring large scale environmental health crises. This provides an opportunity to compare the risk perception of both crises in the population. In particular, whether experiencing the acute pandemic sensitizes people to the risks of ongoing climate change. Methods: Panel participants answered a web-based questionnaire. The risk perception of SARS-CoV-2 and influencing factors were assessed. Differences of risk perception dimensions regarding SARS-CoV-2 and climate change were analyzed as well as associations between dimensions. Results: The results show that an economic impact by the pandemic is associated with more dimensions of SARS-CoV-2 risk perception than an experienced health impact. Moreover, dimensions of risk perception of the pandemic and climate change are perceived differently. Furthermore, the affective dimension of pandemic risk perception is significantly associated with all dimensions of climate change risk perception. Conclusions: Emotional-based coping with the risks of SARS-CoV-2 is associated with risk perception of climate change as well as various factors that shape the individuals’ risk perception. It is currently necessary and will be increasingly necessary in the future to solve coexisting crises, not selectively, but in a common context within the framework of a social-ecological and economic transformation.

## 1. Introduction

Risk research proposes different factors that influence peoples’ risk perception [1]. Because lay-people may have difficulties with the interpretation of probabilities, they tend to use other factors, such as emotions, availability (e.g., current events, media presence), or consequences regarding (taking) the risk (i.e., personal or general) for risk assessment. This may lead to biased risk estimation and also shows the multidimensionality of risk perception [1,2,3]. Currently, significant predictors for a higher risk perception of the SARS-CoV-2 pandemic were identified, such as personal concern or direct and indirect personal experience with the virus, socio-cultural aspects and views, as well as an active confrontation with the current crisis [4,5,6,7]. That also includes pandemic-associated health and financial impacts [8]. Overall, a higher risk perception, as well as the factor *fear*, are positively associated with the adoption of (personal) preventive measures [8]. Studies on risk perception have become increasingly relevant during the SARS-CoV-2 pandemic because adequate risk communication based on this knowledge is an important tool that can help to improve and mediate health strategies as well as to build confidence and contributions to a collaborating governance [4]. It has yet to be elucidated whether, or to what extent, experiencing the acute pandemic makes people aware to the risks of other environmental health crises, such as the ongoing climate change.

Currently, SARS-CoV-2 and climate change are globally simultaneous environmental risks that threaten the health of humankind, but endemics and pandemics will be more likely in the future, driven by multiple environmental factors, such as changing climate and ecosystems [9]. Moreover, the unmitigated proceeding of global climate change increases the probability of several health risks that come along with, for example, extreme weather events, such as heat waves, droughts, heavy rain, and flooding [10]. As a result of these developments, the population will be increasingly affected directly and indirectly by health-related, existential, and economic consequences of environmental health crises. Although people may perceive a general risk of climate change, they often do not associate health consequences with it or they do not feel concerned, because its impacts still occur in temporal and spatial distance [7,11,12,13,14]. Until the time when consequences actually occur, risks such as pandemics and climate change are widely regarded as low probability—high consequences risks [15]. Nonetheless, the existence-threatening consequences of climate change are not only part of a worldwide discussion, but are also of increasing concern of the population in Germany [6].

Both the SARS-CoV-2 pandemic and climate change are very complex in terms of their origins and consequences. Therefore, solutions to problems from past emergency events are hardly transferable [16]. Although both crises proceed on different time scales (i.e., evolving at a different pace), promptly addressing mitigation and adaptation is necessary to prevent the disruption of social systems [16,17]. Impacts (e.g., socioeconomic, health) following this disruption will not only affect vulnerable groups, but all individuals of the global population. In fact, the pandemic reinforces not only the vulnerability of humanity to nature [18], but offers a first view of future distress that may arise due to climate change [16]. Hence, the challenges of the current pandemic may offer a chance to get prepared to handle similar climate change-related situations in the future, e.g., coping with irreversible changes, socio-spatial disparities and polarization of the society, the weakening of global solidarity, and the handling of increasing costs [19]. 

Successful coping with environmental health crises such as the global SARS-CoV-2 pandemic and climate change can only be realized by protection and prevention measures supported by society as a whole. Therefore, it is essential that these (from a scientific perspective) predictable crises are recognized by the population as a health risk as well as an existential threat [5]. There are indications that perceived personal threat due to pandemics, climate change, habitat loss, and extinction is an important factor to increase the individual and population-based willingness of behavioral change that may lead not only to self-protecting behavior but also to a climate-protecting and sustainable lifestyle [20,21]. 

Hence, personal risk perception may be an important precursor to health-related and other behaviors that determines how individuals respond to potential hazards [22]. For example, it has been stated that risk perception of climate change (such as floods or landslides) increases if individuals experience its consequences [14]. The SARS-CoV-2 pandemic is a result of environmental destruction and globalization [23,24], but currently it is not conceivable to what extent the experiences and distress of the pandemic may sensitize people and influence the individuals’ willingness to attain more sustainability and climate protection.

The current study aimed to analyze on a population level (1) whether personal health or economic impacts increase the risk perception of the SARS-CoV-2 pandemic. In addition, we analyzed (2) whether risk perception dimensions of the pandemic and climate change are perceived differently and (3) if dimensions of risk perception of the SARS-CoV-2 pandemic are associated with dimensions of risk perception of the simultaneous crisis of climate change.

## 2. Materials and Methods

### 2.1. Study Design

This cross-sectional study was carried out in February 2021, marking the second lockdown in Germany to manage the spread of SARS-CoV-2. Participants were recruited in North-Rhine Westphalia (NRW) in cooperation with the Institute for Sociology and Communication (SOKO Institute) via an ISO-certified Access Panel of the company *Respondi* (www.respondi.com, accessed on 15 January 2021). The panel operator sent invitation links to the participants and paid them a small financial incentive after completing the survey. The quota sampling was based on socioeconomic variables (i.e., age, gender, net household income, and size of residence) according to the distribution in NRW [25]. A sample size of *n* = 1000 was planned and participants were recruited until all quota for the sample were fulfilled. The questionnaire was created by the authors of this study within the online survey tool *LimeSurvey* (www.limesurvey.org, accessed on 15 January 2021). A questionnaire with closed items was developed in German and assessed questions such as sociodemographic characteristics, health and economic impacts of SARS-CoV-2, and risk perceptions of SARS-CoV-2 and of climate change. The questionnaire was pre-tested with *n* = 100 participants. Data were analyzed anonymously. The study was approved by the Ethics Committee of Bielefeld University (No. 2020-205).

### 2.2. Study Population

The raw data sample comprised *n* = 1160 participants. After excluding participants due to not fitting the target population (i.e., not living in NRW; *n* = 37), incomplete questionnaires (*n* = 49), and response time less than 6 min (2 s/item), which is considered insufficient effort responding [26] (*n* = 25) the study sample comprised *n* = 1049. Additionally, for the analysis of risk perception, a test of consistency was conducted. Participants who disregarded reverse-coded questions on the risk perception of the SARS-CoV-2 pandemic or climate change, and thus provided inconsistent responses, were not included in the analyses of risk perception. As a result, the sample size was reduced to *n* = 918 (Figure 1). This sample reduction did not affect the previously set quotas. 

### 2.3. Measures

#### 2.3.1. Sociodemographic Characteristics and COVID-19 Risk Factors

The first part of the questionnaire included questions about the socioeconomic status of the participants (age, sex, net household income, educational status, and residence size). Moreover, the presence of pre-defined risk factors [27] for the severe progression of COVID-19 (cardiovascular diseases, diabetes, diseases of liver, respiratory system or kidneys, cancer, obesity, smoking, or suppressed immune system) was assessed. For the analyses, the variable was dichotomized into *no risk factor* and *one or more risk factor(s)*. Answers regarding risk factors were tested for plausibility.

#### 2.3.2. Health and Economic Impact

Health impact was measured as direct or indirect impact. A person was considered directly affected if he or she contracted SARS-CoV-2 before the survey. Additionally, the progression of COVID-19 was asked (i.e., without/with mild/severe symptoms, hospital admission, Intensive Care Unit (ICU) admission). If participants stated they had persons close to them who have been infected with SARS-CoV-2, they were considered as having an indirect health impact. Additionally, they were asked how the disease proceeded. As the question was created with multiple choices, a plausibility analysis was conducted. For the analyses, direct and indirect health impacts were dichotomized into *affected* and *not affected*. Furthermore, participants were asked whether they were affected economically by the SARS-CoV-2 pandemic or worried about economic consequences. Questions were adapted from other surveys regarding the same topic [28,29]. The questions include personal concern about economic consequences of their own household (from 1 ‘not concerned at all’ to 5 ‘very concerned’) and actual financial losses (from 1 ‘no loss at all’ to 6 ‘very large losses’). In addition, participants were asked whether and to what extent their occupational status had changed during the pandemic (‘no’; ‘yes, I am currently in short-time work’; ‘yes, my working hours have been contractually reduced’; ‘yes, I have lost my job’; ‘yes, I am a freelancer/self-employed and I am currently not able to do my job as usual or receive less orders than usual’). The first two questions of economic impact were recoded to scales ranging from 0 to 10 and aggregated into one dichotomized variable where values below the mean indicated *low economic impact* and values above the mean indicated *high economic impact*.

#### 2.3.3. Risk Perception

Risk perception of the SARS-CoV-2 pandemic and climate change was measured according to the risk perception scale developed by Wilson et al. [3]. The multidimensional tool was specifically developed to measure the risk perception of various hazards [30]. The tool includes three dimensions, namely *affect*, *probability*, and *consequences*. *Affect* refers to the emotional handling of the risk (five items), *probability* to the assessment of how probable the risk affects oneself or the community (three items), and *consequences* to the perceived severity of a risk or hazard (two items). The validation conducted by Wilson et al. [3] included similar risks (i.e., extreme weather events (climate change) and food poisoning (health-related)), which makes it a useful tool for this study as well. For application, the authors translated the instrument into German. To ensure that the translated items still represent the core dimensions of the original scale, the back-translation method was used. Items were slightly adapted regarding the corresponding hazard (i.e., SARS-CoV-2 and climate change; (Table 1)). 

All items were answered via 5-point-Likert-scales (1 = ‘not at all’ to 5 = ‘very much’). Values for each dimension (affect, probability, consequences) were summarized by calculating means of the individual items. Higher values represent higher risk perception. The internal consistency (based on Cronbach’s alpha values) of the three scales for the SARS-CoV-2 pandemic (affect: 0.91, probability: 0.82, consequences: 0.81) and climate change (affect: 0.91, probability: 0.81, consequences; 0.83), respectively, was found to be good. One item of the probability dimension asking “How often do infections with SARS-CoV-2 occur at your place of residence?” was excluded. Due to the high dynamics with steadily increasing infection numbers during the second lockdown in Germany, the question was not purposeful to depict a differentiated assessment of the (subjective) estimation of infection spread.

### 2.4. Statistical Analysis

To assess associations between health impact or economic impact and SARS-CoV-2 risk perception dimensions, individual linear regression models were conducted for each dimension (adjusted for age, sex, education, risk group). Differences in the dimensions of risk perceptions of SARS-CoV-2 and climate change were tested via paired t-tests.

The association between risk perception of the SARS-CoV-2 pandemic and climate change was tested via three individual linear regression models. Each regression model had one dimension of risk perception of climate change as the outcome variable (i.e., *affect, probability, consequences*) and all three dimensions of risk perception of SARS-CoV-2 as the explanatory variable. The models also controlled for the possible confounding variables of age (continuous), sex (dichotomized), education (categorial), and health risk factor for severe COVID-19 progression (dichotomized). Data cleaning and analysis were conducted with IBM SPSS Version 28 and the R Statistical Environment [31] with ‘tidyverse’-package [32] and ‘psych’-package [33]. Statistical significance was considered a *p*-value of up to 0.05.

## 3. Results

### 3.1. Sample Characteristics

The composition of the study sample (*n* = 1049) is representative for North-Rhine Westphalia, a German state in Western Germany, due to the quota recruiting method (Table 2). Participants’ (male: *n* = 523; female: *n* = 524; divers: *n* = 1) ages ranged from 16 to 85 years (mean = 48.4, SD = 17.3). Females with an average age of 44.2 years (SD = 17.9) were slightly younger than males with an average age of 52.7 years (SD = 15.6). Net household income was almost equally distributed between the categories, except for the income group ‘less than 1000 €’ (EUR 1000), which only accounts for 7.2% of the sample corresponding to the NRW quota of 7% (Table 2).

More than half of participants (53.6%, *n* = 560) stated having at least one risk factor for severe progression of COVID-19. One third of the participants (31.6%, *n* = 330) stated having pre-existing condition(s) (cardiovascular, diabetes, diseases of liver, respiratory system or kidneys, cancer) and a quarter stated being a smoker (25.9%, *n* = 270). The risk factors obesity and suppressed immune system were less common (12.6%, *n* = 131; 5.8%, *n* = 60, respectively) in the sample. 

### 3.2. Health and Economic Impact by the Pandemic

#### 3.2.1. Health Impact

At the time of the survey (February 2021) most of the participants (91.5%, *n* = 956) stated that they had not been infected with SARS-CoV-2. The participants reporting an infection (8.5%, *n* = 89) had mostly no symptoms (60.7%, *n* = 54) or light symptoms (23.6%, *n* = 21). Only a few participants reported hospital (3.4%, *n* = 3) or ICU admission (1.1%, *n* = 1), while 11.2% (*n* = 10) reported severe symptoms without hospital admission.

An indirect health impact (i.e., infection of one or more close persons; multiple response option), was reported by 37.7% of participants (*n* = 361). Of those reporting an infection of close persons, 12.2% (*n* = 127) passed their infection without symptoms, and 17.8% (*n* = 185) with light symptoms. Less common were severe symptoms (9.3%, *n* = 97), hospital (3.0%, *n* = 31), or ICU admissions (1.3%, *n* = 13). Several participants (1.6%, *n* = 17) reported the death of a close person due to the COVID-19 disease. After dichotomization, the ratio of health impact resulted in 63.1% not affected and 36.9% affected participants in the study population.

#### 3.2.2. Economic Impact

The economic impact of the SARS-CoV-2 pandemic was evaluated, including whether changes in the occupation status occurred. The majority of participants (82.3%, *n* = 857) answered that there was no change. However, some participants reported that they were either temporary part-time (6.9%, *n* = 72), lost their jobs (4.2%, *n* = 28), were freelancer/self-employed and could not do their occupation as usual (3.9%, *n* = 44), or had a contractual reduction in working hours (2.7%, *n* = 41).

Participants answered regarding their concerns of personal economic impact heterogeneously. More than half reported being little (25.5%, *n* = 266) to moderately (26.2%, *n* = 274) concerned. Only 18.2% (*n* = 190) were not concerned at all, while 17.3% (*n* = 181) were rather concerned and 12.8% (*n* = 134) were very concerned.

Most of the participants (44.1%, *n* = 463) had no financial losses. Yet about 28% had very little (10.9%, *n* = 114) to little (17.3%, *n* = 181) financial losses. Large (6.7%, *n* = 73) to very large (4.0%, *n* = 42) financial losses after almost one year of the SARS-CoV-2 pandemic were less common. After dichotomization (*low* and *high* economic impact), the ratio in the study population was 51% low and 48% of a high impact.

### 3.3. Associations between Health/Economic Impact and Risk Perception of SARS-CoV-2

The association between health or economic impacts and risk perception of SARS-CoV-2 were analyzed via individual linear regression models. There was a significant association between health impact and the probability dimension of risk perception of SARS-CoV-2 (B = 0.29, *p* < 0.001). Thus, participants who reported a health impact because of the pandemic, also perceived an infection with SARS-CoV-2 as a higher risk (Table 3). A higher economic impact is associated with the affect dimension, but also with the probability dimension of risk perception (Table 3). Neither a health impact nor an economic load had an effect on the dimension *consequences* concerning a SARS-CoV-2 infection. 

### 3.4. SARS-CoV-2 Risk Perception and Climate Change Risk Perception

Regarding the mean values of the risk perception of the pandemic and climate change in the dimensions *affect* and *consequences* (i.e., how concerned people are about one risk and how their individual impact would be if being infected or affected by climate change, respectively), significant differences between both regarded risks are noticeable (Table 4). In both dimensions, *affect* and *consequences*, participants perceived the pandemic as riskier (affect: SARS-CoV-2: mean = 3.42; climate change: 3.31; *p* < 0.05/consequences: SARS-CoV-2: mean = 3.47; climate change: 3.30; *p* < 0.05) (Table 4). The perception of the probability that one of the risks will occur was also significantly different between both risks, whereas participants assessed the probability of the occurrence of climate change (mean = 3.31) as higher than the probability of contact with a person tested positive for SARS-CoV-2 (mean = 2.46, *p* < 0.001).

Possible associations between single dimensions of risk perception of the two environmental health crises were analyzed. All models were controlled for age, sex, education, and risk factors (Table 5). The models show an association of the *affect* dimension of SARS-CoV-2 with the corresponding outcome dimensions *affect*, *probability*, *consequences* of climate change risk perception. Moreover, an increase in the dimensions of *probability* and *consequences* of a SARS-CoV-2 infection is associated with an increase in the *probability* and *consequence* dimension of climate change risk perception, respectively (Table 5).

## 4. Discussion

The aim of the current study was firstly to analyze the risk perception of the SARS-CoV-2 pandemic in Germany and whether an impact (health or economic) by the pandemic may influence the risk perception of SARS-CoV-2. The study reveals that economic impacts are associated with more dimensions of risk perception of the SARS-CoV-2 pandemic than health impacts. Secondly, it has been assessed if dimensions of risk perception of the pandemic are associated with dimensions of risk perception of the current climate crisis. We could show that especially the affective dimension of risk perception of the SARS-CoV-2 pandemic is associated with all dimensions of risk perception of the currently occurring climate change. The unique situation of the parallel occurrence of two environmental health crises offered also the opportunity to analyze whether the respective risks are perceived differently. Regarding affect and consequences dimensions, SARS-CoV-2 risk perception is higher than climate change, whereas the probability dimension is rated higher for climate change.

### 4.1. Risk Perception of SARS-CoV-2 and the Influence of Health and Economic Impacts

The SARS-CoV-2 pandemic had a massive impact on the lives of many people in Germany, especially with regard to being affected by health or economic consequences due to the infection events and protection measures. 

The present study shows that the personal experience of health consequences due to the pandemic, i.e., their own infection or of people in their close social environment, contributes to the rating of the probability of becoming infected as higher. In contrast, no association was found between a personal health impact and how severe the consequences due to a SARS-CoV-2 infection are estimated to be. According to another study, the “experience with the virus”, i.e., having tested positive for SARS-CoV-2, was identified not to be the primary but the third most important predictor for risk perception in various countries, after individualistic views and pro-sociality [5].

The higher risk perception of SARS-CoV-2 in the dimension of *probability* may be related to the time point of measurement. Hetkamp et al. [34] found in their study that in the first weeks of the pandemic, fear of SARS-CoV-2 increased in concordance with infection numbers. Another study also showed that the general SARS-CoV-2 risk perception was significantly different over time and in line with occurrences due to the pandemic, i.e., lockdowns, new variants, restrictions, infection numbers [35]. 

Therefore, when estimating risk perception, the time point of measurement or the current context of threat is important for a proper interpretation, respectively. There are indications that the temporal nature of the consequences certainly impacts how people place weight on different risks [3]. In the case of the pandemic, consequences occurred immediately for many people, whereas consequences of climate change may occur in a delayed manner and are less certain, which is why they are more often perceived as being rather minor in the present [3].

With regard to personal economic impacts due to the pandemic, we could reveal that a higher economic impact (i.e., worries about and actual financial losses) was significantly associated with an emotional involvement concerning worries and fears related to the pandemic, operationalized by the risk perception dimension *affect*. Moreover, in accordance with health-affected people, economically affected participants also estimated the *probability* for getting infected or having contact with an infected person as higher than people with low economic impact. Overall, the results indicate that economic worries due the pandemic influence people more than health worries. This assumption is supported by the findings of a worldwide study that considered what kind of concerns due to SARS-CoV-2 are higher and predict self-protective behavior better [36]. Consistent in all included countries of the study was that people perceived economic risks higher than health risks due to SARS-CoV-2. Additionally, perceived economic risk was found to better predict all kinds of preventive behavior [36]. The importance of the economic impact factor in influencing perceptions of pandemic risk can be explained, to some extent, by the fact that during the survey period, the probability of suffering economic losses from the pandemic was estimated to be nearly half of the global labor force (in 2020). Instead, the probability of becoming infected with the virus was considered to be low to moderate for the general population [36]. Therefore, economic impact is perceived as a higher personal and existential risk, and brings uncertainties that accordingly trigger a higher emotional impact, but interestingly, Nisa et al. [36] reported also that people do not perceive health and economic risks as competing with each other.

Although it has also been stated that emotions promote risk perception often more effectively than knowledge and facts [5,8], it is beneficial for a holistic approach to address knowledge and facts as well as emotions as an adequate risk communication strategy. Hence, for risk communication that intends to promote risk awareness and understanding, reducing risky behavior, or encouraging risk-minimizing behavior, all these aspects should be addressed. This applies both to the communication concerning the health consequences of the SARS-CoV-2 pandemic as well as risks such as climate change.

Thus, to successfully manage an environmental health crisis, risk communication strategies and crisis management must consider the potential consequences (e.g., health or economic) and perceptions of risk, as well as the necessary preventive behaviors.

### 4.2. Risk Perception in Contemporaneous Environmental Health Crises

Survey participants rated the emotional risk dimension and consequences of the pandemic higher than those of climate change, possibly because an immediate threat was perceived to be more present. Instead, the dimension *probability* with regard to getting infected was lower than the estimated probability of being affected by climate change. It is conceivable that contact with an infected person is more likely to be considered avoidable to lower the probability to infect yourself, while the probability to be affected by climate change is less controllable and cannot be prevented by only one’s own decision or behavior [12]. 

Moreover, the analysis of the risk perception of the two parallel occurring health-related crises, the SARS-CoV-2 pandemic and climate change, showed that the risk perception dimensions *affect, probability,* and *consequences* of both crises are associated. In particular, the emotional risk perception of the SARS-CoV-2 pandemic (*affect* dimension) showed statistically significant associations with all dimensions of risk perception regarding climate change.

This result indicates that being emotionally affected by a crisis such as the SARS-CoV-2 pandemic might have an impact on the risk perception of another crisis, such as climate change. In particular, a negative frame of information captures the attention of people and supports negative emotions that may sensitize people to otherwise neglected risks for themselves or others [8]. Similar results were found in a study from Mohommad and Pugacheva [18], which investigated the impact of SARS-CoV-2 on attitudes to climate change and support for climate policies polled in 16 countries with 14,514 respondents aged 16–74 years. They identified that experiencing a direct or indirect SARS-CoV-2 health shock has a positive and significant correlation with the likelihood of being more worried about climate change since the pandemic [18].

Risk perception of a current crisis is, in addition, an important determinant of planned and actual self-protective behavior [37]. The threat experienced by the SARS-CoV-2 pandemic could, therefore, possibly raise awareness in the population also to the (health) risks and consequences of climate change. This assumption is supported by the fact that the dimensions *probability* and *consequences* of the pandemic risk perception are associated with the same dimensions of climate change. It has to be analyzed in further studies if this awareness also promotes preventive behavior.

Also in line with the study results regarding the influence of being affected by the SARS-CoV-2 pandemic is a study by Kecinski et al. [38], which found that people who were very concerned about spreading the virus by themselves were more concerned about environmental protection overall [38]. The same study also found that people whose food supply was limited for one or more days due to their economic impact by the SARS-CoV-2 pandemic also reported greater concern about environmental protection than people who were not affected [38]. Thus, existential worries trigger stress and emotional reactions and, therefore, make those affected more sensitive to other environmental issues and social problems. 

There is further evidence that the co-occurrence of another risk, or a higher risk perception of another hazard, leads to threats such as climate change being taken more seriously as a global environmental health risk [39], but, the high complexity of the relationship between environment and health, in addition to the lack of knowledge and understanding, can contribute to uncertainties regarding individual risk assessment and of possible consequences of climate change. There are also indications from the literature that people may have an ambivalent attitude towards the existence and extent not only of the current pandemic [40] but also of climate change, since the scientific evidence seemed unreliable, insufficient, or contradictory [12,41]. In addition, political and social controversies in dealing with the consequences of crises such as the pandemic and climate change further strengthened the insecurity [12]. Accordingly, the link between environmental risk factors and human health needs to be made more explicit in risk communication and climate change policy, particularly by providing reliable information based on sound knowledge.

Future studies should examine more closely whether being affected by a crisis such as the pandemic will increase the awareness of the simultaneous climate change. In particular, whether being affected might promote a positive spillover effect, in the sense that experiencing one event has a positive impact, i.e., on attitudes concerning another event that is independent of it [42,43]. That also includes an assessment on to what extent a possible pandemic impact may motivate people to self-protective or pro-environmental behavior, respectively. 

### 4.3. Implications for Risk Communication in Environmental Health Crises

The present study supports the assumption that a multidimensional measurement of risk perception might be a suitable tool to map the facets of risk perception in a more differentiated way, because the assessment of risks is individual and influenced by many factors. Therefore, the development of appropriate risk communication strategies continues to be a challenge. With regard to coping with environmental health crises, this and other studies have shown that emotions, especially as a result of personal concern, can significantly increase risk perception [5,38,44]. For a successful risk communication, it is, therefore, particularly important to consider the needs of different risk groups in their current context. Moreover, it is necessary to reach people who are not (yet) affected by the consequences of a crisis and motivate them to take preventive action. The goal of further risk research should therefore be, in particular, to improve the transferability of findings from one particular risk context to another, such as from the pandemic to future consequences of the climate crisis. Additionally, proper risk communication should aim to mediate appropriate knowledge to encourage the public to take action [45] and promote an emotional involvement to support peoples’ willingness for behavioral change. However, it must also be stated that it is challenging to counter mistrust and skepticism through convincing risk communication [46] and to reach all (sub)groups of society, especially by addressing emotional involvement.

## 5. Limitations

The results of the present research are limited in some respects. Inherent to cross-sectional study designs, the possibility to draw causal inference is restricted. Moreover, the results rely on self-report measures, which could have caused the results to be contaminated by common method variance. While this is the usual practice for most of the examined constructs, concerns about the validity of self-reports cannot be excluded [47]. The present study investigated the situation towards the end of the second wave of the SARS-CoV-2 pandemic in Germany. The unique situation of a dynamic pandemic limits the survey period, and findings here can only be a snapshot, so further investigations could be worthwhile. 

Wherever available, we used validated measures. However, due to the novelty nature of the pandemic, it was not possible to use validated instruments at all points; therefore, a precise measurement of certain study constructs might be limited. Accordingly, future research should aim to verify the results of this study, as well as validate, and incorporate further measurements. 

In the study, the instrument developed by Wilson et al. [3] was used to capture and compare the multidimensional perception of different risks. However, the results for different risks have to be considered very differentiated with regard to the respective context, because they seem clearly to depend on individual affective responses to the respective risk [3]. In addition, as already has been stated by Wolff et al. [41], when constructing questionnaires, it is of great importance to be aware of possible biases, “otherwise methodological artifacts resulting from item wording might be misinterpreted as substantial findings”. For example, during periods of pandemic with high incidences, the question on the dimension *probability* (“How often do infections of SARS-CoV-2 occur where you live?”) can only be answered with minor differentiation, and thus only allow an imperfect assessment of the risk perception.

Considering this, follow-up studies are recommended that incorporate further informants and other methods of data collection (e.g., experience sampling/ecological momentary assessment (EMA), in-depth qualitative methods) to obtain a more comprehensive picture. In particular, longitudinal studies should be carried out to further substantiate the evidence for the large effects found. Additionally, different approaches of sample recruitment to ensure a broader representation of the population might be considered in future research. 

## 6. Conclusions

Emotional-based coping (i.e., high affective risk perception) with the risks of SARS-CoV-2 is associated with risk perception of climate change as well as various factors (e.g., health, economy, world views) that shape the individuals’ risk perception. Hence, successful climate protection is an economic as well as an ecological and social challenge. It is currently necessary, and will be increasingly necessary in the future, to solve coexisting crises, as well as pandemics, not selectively, but in a common context within the framework of a social-ecological and economic transformation. This requires a better understanding not only of what people know about crises such as climate change and the SARS-CoV-2 pandemic, but also of the experiential, social, and cultural factors that shape risk perceptions and their potential role in motivating preventive behavior. Better insight will help policymakers and other stakeholders develop evidence-based risk communication strategies tailored to societal subgroups.

## Figures and Tables

**Figure 1 ijerph-20-03363-f001:**
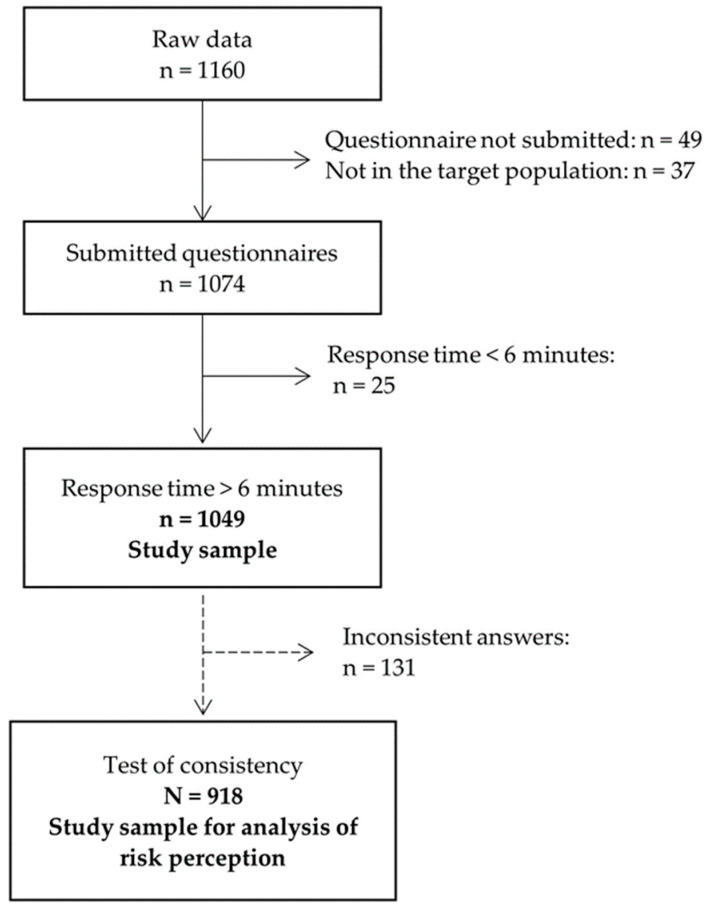
Sample size flow chart within data cleaning process.

**Table 1 ijerph-20-03363-t001:** Dimensions and related items of the risk perception scales for SARS-CoV-2 and climate change (based on Wilson et al. 2019 [3]).

Dimension	Item
Affect	How concerned are you about SARS-CoV-2 pandemic/climate change? When you think about SARS-CoV-2 pandemic/climate change for a moment, to what extent do you feel fearful? When you think about SARS-CoV-2 pandemic/climate change for a moment, to what extent do you feel anxious? When you think about SARS-CoV-2 pandemic/climate change for a moment, to what extent do you feel worried? Considering any potential effects that SARS-CoV-2 pandemic/climate change might have on you personally, how concerned are you about SARS-CoV-2 pandemic/climate change?
Probability	How likely is it that you will have contact with someone who has tested positive for the SARS-CoV-2 within the next four weeks/How likely is it that consequences due to climate change will occur where you live in the future? I am confident that I will not have contact with someone who has tested positive for the SARS-CoV-2 within the next four weeks/I am confident that consequences due to climate change will not occur where I live in the future. How often do infections of SARS-CoV-2 occur where you live/Will consequences due to climate change occur where you live in the future?
Consequences	If I were to be infected (again) with SARS-CoV-2 it would probably affect me negatively/If I did experience consequences due to climate change it would probably affect me negatively. If I were to be infected (again) with SARS-CoV-2 it would likely have negative impacts on me/If I did experience consequences due to climate change it would likely have negative impacts on me.

**Table 2 ijerph-20-03363-t002:** Sample characteristics (*n* = 1049) compared to the distribution in NRW *.

	*n*	%	% NRW Distribution *
Sex	1048		
Female	524	50.0	51
Male	523	49.9	49
Diverse	1 **	0.1	
Age (in years)	1048		
16–29	209	19.9	21
30–39	142	13.6	14
40–49	169	16.1	16
50–59	212	20.2	19
60–69	167	15.9	15
≥70	149	14.2	15
Net household income	996		
<EUR 1000	72	7.2	7
EUR 1000–2000	233	23.4	24
EUR 2001–3000	248	24.9	27
EUR 3001–4000	207	20.8	20
≥EUR 4001	236	23.7	22
Size of residence (inhabitants)	1037		
<100.000	283	27.3	27
100,000–499,999	350	33.8	33
≥500,000	404	39.0	40

* NRW distribution [25]. ** Only one person indicated diverse as sex and could not be included in the analyses as an individual case.

**Table 3 ijerph-20-03363-t003:** Results of the six individual linear regression models regarding the association of pandemic-related health or economic impact and SARS-CoV-2 risk perception dimensions.

Pandemic Impact	Dimensions of Risk Perception	N	B	CI	*p*
Health (yes/no)	Affect	899	0.11	−0.01–0.24	0.083
	Probability	898	0.29	0.16–0.42	<0.001
	Consequences	897	0.05	−0.09–0.19	0.454
Economic (low/high)	Affect	896	0.45	0.32–0.57	<0.001
	Probability	895	0.26	0.14–0.39	<0.001
	Consequences	894	0.12	−0.02–0.26	0.087

N = observations included in regression; B = beta-coefficient; CI = 95% confidence interval; *p* = *p*-value (values < 0.05 are in bold); adjusted for age, risk group, sex, education.

**Table 4 ijerph-20-03363-t004:** Means of the dimensions of risk perception of the SARS-CoV-2 pandemic and climate change.

	Pandemic Mean (SD)	*n* ^1^	Climate Change Mean (SD)	*n* ^1^	*p* ^2^
Affect	3.42 (0.96)	916	3.31 (0.98)	918	<0.05
Probability	2.46 (0.97)	915	3.31 (0.93)	910	<0.001
Consequences	3.47 (1.07)	914	3.30 (0.98)	917	<0.001

^1^ *n* = valid answers (i.e., without missing data); ^2^ *p*-value for paired t-test, SD = standard deviation.

**Table 5 ijerph-20-03363-t005:** Results of the three adjusted linear regression models regarding the association of all dimensions of risk perception.

Dimensions of SARS-CoV-2 Risk Perception ^1^	Dimensions of Climate Change Risk Perception *^2^*	N	B	CI	*p*
Affect	Affect	895	0.33	0.26–0.41	**<0.001**
Probability			−0.02	−0.08–0.05	0.592
Consequences			0.00	−0.07–0.07	0.992
Affect	Probability	887	0.21	0.13–0.28	**<0.001**
Probability			0.07	0.01–0.13	**0.031**
Consequences			0.04	−0.03–0.10	0.301
Affect	Consequences	894	0.20	0.12–0.27	**<0.001**
Probability			0.05	−0.01–0.12	0.087
Consequences			0.20	0.13–0.26	**<0.001**

^1^ predictor; ^2^ outcome; N = observations included in regression; B = beta-coefficients; CI = 95 % confidence interval; *p* = *p*-value (values < 0.05 are in bold); adjusted for age, risk group, sex, education.

## Data Availability

Data are available on request by contacting the corresponding author.

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
