# Peer review of "Risk Perception of the SARS-CoV-2 Pandemic: Influencing Factors and Implications for Environmental Health Crises"

_ijerph, 2023, doi:10.3390/ijerph20043363_

Round 1

Reviewer 1 Report (Previous Reviewer 1)

This iteration is very much improved with a consistent thread, making it an enjoyable read. With a few alterations in places, I'm happy to suggest it is accepted for publication.

Some points for amendment (some just text alterations - I picked them up so thought I might as well include if it helps in some way - some are just comments). Not all require action.

40: “fear factor”?

44: comma in middle of sentence not needed

45: “build confidence”?

46: “extent”?

79: would “foreseeable” be a better term than “predictable”?

86: “e.g.” – could this be worded slightly better?

92: upper case needed

258: should this be SARS-CoV-2? Few of these dotted about.

269: I think affect and consequences are italicised elsewhere?

313: sentence needs rewording for clarity

317: “testing”?

318: should that be “behind” rather than “besides” – i.e. those other two predictors come before testing positive?

339: “indicated” rather than “could indicate” – at times the writing style could use fewer qualifiers

358: “intends”?

355: just as a more general comment that may or may not be of interest: more recent work in the sociology of emotions looks to reinterrogate how we understand the relationship between emotion and reason (“knowledge and facts”), seeing them not as binary, but with each as necessary for the other. Doesn’t require action, just made me think.

369: remove “the” before climate change

423: “extent”?

427, 432, and 435 below are connected

427: I think you may push to heavily on the individuality of risk assessment when you are discussing throughout how this may be patterned in some way. Maybe tone this down slightly in this opening sentence, perhaps just by removing the “very”?

432: Similarly, what are “people’s individual needs”? Do you mean that the variation in contexts people find themselves in leads to the need for heterogeneity in communication? If so, then there are still these broader patterns, and not so much individual needs. Perhaps a slight rewording again here? This would certainly help to translate this across disciplines too.

435: in relation to the last two points, you go a long way toward this here, by talking about risk contexts (much more in line with what your paper has done up to this point) – maybe use this language more in the two points above.

Author Response

Reviewer 2 Report (Previous Reviewer 2)

Thank you for addressing the comments that were previously raised.

Author Response

This manuscript is a resubmission of an earlier submission. The following is a list of the peer review reports and author responses from that submission.

Round 1

Reviewer 1 Report

Overall feedback

Overall, the article contributes in an important research domain with potential practical consequences; i.e., the realisation of communicative opportunities across crises. The article also presents some interesting findings, which have interdisciplinary relevance for studies of climate change communication. However, in my opinion, more clarity is required in a number of places before this paper can be considered for publication.

The case for researching in this area, and for those that have already done so, I feel, needs to be made more powerfully in the introduction, to build a fuller picture for the reader. The authors build on such research in the discussion, some of which could have been used to build this picture in the introduction. More justification for the health and economic impacts hypothesis could also be provided (based on prior research), perhaps via an additional paragraph in the introduction.

The method appears to me to be strong, though I’m not convinced of the explanation (and therefore justification) for indirect health impacts and the measure constructed around it. Perhaps including the (translated) survey questions themselves would be preferable here, or maybe including them in a supplementary file. If not this, then a more focused and clear explanation of what this is actually looking to measure is required.

In the discussion some really important points are made, but at times it drifts too far away from the present study and focuses on other research in a way too far removed from its relevance to the data. Again, some of this may well have been included in the introduction to anchor this work, and thus avoid having to go into too much detail away from the work/data itself. Perhaps more references back to what this study’s analysis means in conjunction with that literature would help as a guiding thread throughout the discussion, particularly as, at times, it seems a little ‘scattershot’ or disorganised.

Despite a fairly small sample size, I think the method and results here add an interesting contribution to the literature, but, personally, I would want to see (and would be happy to see) a revised paper first with greater clarity and more of a thread tightening the piece up as a whole (from introduction to discussion). Some spell- and phrase-checking might also help in this regard (i.e., tightening up the work), as in places clarity is lost due to disconnected sentences and choice of words.

More specific feedback:

Abstract:

·       “Provides opportunity”…  but why is this important? For what can be gleaned for climate change or environmental communication more generally? 13

·       “Depending on the nature of personal concern” – isn’t entirely clear, what is meant by personal concern, exactly? Is there a better – though still short – explanation for this? 22

·       The abstract reads as if words (particularly definite articles, i.e. “the”) have been removed to reduce word-count.

·       The final two sentences are hard to follow: can slightly more information be given about the “emotional dimension”, and is “economic stability” at the personal or the system level?

References

·       Is there a reason that citations are in different formats?

Introduction

·       “Barely transferable” (40-41) – a very short sentence which doesn’t immediately make a great deal of sense. “Other crises”? As in not COVID or climate change? This could be clearer.

·       The third paragraph (40- ) relies very heavily on Engler et al. (2020), is there any other literature that also compares COVID and climate change?

·       “Risk research proposes different assumptions of how people perceive risks” (66): different assumptions from what? What is it being compared with here?

·       Different assumptions to what?

·       Laymen – perhaps use of lay-people or lay perceptions would be more suitable? (67)

·       “(e.g., emotions, availability, consequences)” (68): I understand what “emotions” is doing here, but availability and consequences could relate to many different things, so more explanation or a different term needed. Later – from line 150 – you outline this but here it assumes prior knowledge of the reader.

·       Is there any literature around the complexities of researching risk perception, or the lack of consensus around a construct? “Highly researched” but nothing cited (69-70). You do go on to cite Wilson et al., - perhaps these two sentences could be merged and made more concise?

·       Again, more information needed about what you mean by consequences (71).

·       Why the hypothesis around health or economic impact (80-82)? Evidence for such a hypothesis, I feel, needs to be bolstered in the introduction for the hypothesis to hold weight.

·       Given the title and thrust of the piece, should “we analyzed whether the dimensions of risk perception of climate change are influenced by the risk perception of the SARS-CoV-2-pandemic” (82- ) not be your primary aim?

Methods

·       Was there some process or discussion about the 6 minutes – how was that decision made? (~106)

·       COVID-10 (117)]

·       Construct of “indirect health impact” is not clear (129- ). It might be easier to directly translate the survey questions used in this section, for reader clarity.

·       Is there a reason why the CFA data aren’t shown? (160-161)

·       Table 1 is unclearly formatted in places – i) is centred text the best option? ii) would lines between groups of questions help interpretation of the text? Otherwise a very useful table.

Results

·       “Most of the participants” (205) – is 31.6% “most”?

·       From 213 onwards, are you talking about the sample of those with infection? This could be clearer (e.g., “Of those reporting infection, …). Same with indirect health impact paragraph (217- ).

·       Wording: “significantly differed”? (245-246 and 249-250)

Discussion

·       Some wording issues that I highlight because they affect the potential meaning of the text:

o   303: would remove “could”

o   312: wording: “the same results” or similar findings

o   340: higher than health risks?

o   343: “could identify” – do you mean there is research that does this, or that this is gap that could be explored?

·       345-346: perhaps a little too certain based on a single study.

·       347 onwards: this is a discussion of literature around the risk perceptions of COVID – more could be done here to draw this back to your findings, or make it more clear that you are doing this. When reading this section I get the feeling it should be in the introduction/literature review.

·       353-358: The first sentence here makes an important divergence or distinction between knowledge/facts and emotion, as does a great deal of research on climate change communication. The second sentence appears to undo that distinction, and doesn’t seem to say a great deal. Also, I know there is an “implications” section, but what does “addressing” these things look like? Does it look different for knowledge/facts and emotions? Perhaps this is just an issue of clarity and drawing back to how it relates to the analysis/results would help.

·       If Mohommad and Pugacheva’s study looks at a very similar topic, should it not have been in your introduction?

·       417: considered by whom?

·       In the implications for the future section, the text appears to largely talk about the implications for the future from other work, i.e., around the importance of “sound knowledge”, though there is emphasis beforehand on the importance of emotional relations emerging in the analysis. Is the connection here, given that it refers to “high complexity of these relationships”, that the emotional dimension impacts on the capacity to develop sound knowledge? If so, this seems an important point and is could certainly be regarded as implications from this study’s analysis, but could be clearer.

·       The initial or key findings reported I think are important but the following narrative in these sections seem to be a little “scattershot” and could perhaps do with some refining for clarity, particularly in how the literature relates to the findings themselves. For example, in both 4.1 and 4.2 the findings are clearly outlined at the top of the sections, but by the bottom it is less clear as to why the information is being provided. Perhaps this could be dealt with largely by including a more instructive writing style (i.e., by making links back to the data throughout or wherever relevant).

Reviewer 2 Report

The study aims to simultaneously explore factors related to COVID-19 pandemic and climate change risk perception. A cross-sectional study was conducted in Germany. Findings suggest that perceptions of COVID-19 and climate change risk are associated and that economic impacts exerted a greater influence on risk perception than health for COVID-19, though the implication of this are not clear.

Introduction

·         Introduction needs to be restructured and addition detail added. The connection between COVID and climate change has not been discussed convincingly, which weakens the justification for examining them together. There is also a lack focus around how examining both could specifically complement each other. It is not clear what this study hopes to accomplish.

·         I am also confused about the aim of the study, which includes COVID-19 and climate change risk perception, but only the health and economic impacts on risk perception for COVID-19?

Materials and Methods

·         What kind of incentive was offered?

·         How were the participants recruited? What was the response rate? What is the quota recruiting method?

·         It is stated several times that consistency was assessed. I think a question is how the survey was validated. What validation measures were undertaken? What does partially validated mean?

·         It seems that certain questions focused on the health and economic questions impacts of COVID-19 only. What about health and economic questions related to climate change?

·         Can the findings from the confirmatory factor analysis be included as supplementary material?

·         I am not clear about the meaning of “…three dimensions (affect, probability, consequences) were joined via mean calculation”. Does this indicate that the mean of all three were taken? Why?

·         I am not clear about the measures of internal consistency (lines 168-171). Do these values (>0.9 and >0.8) represent both risk perception scales for COVID-19 and climate change? Could exact values be provided?

·         I am not also clear how the multivariate multiple linear regression modelling was done. I would have expected the dimensions of risk perceptions for both COVID-19 and climate change to be dependent variables. However, it seems one set was treated as the dependent and the other as independent, which is confusing. How can the risk perception scores of one set be a predictor of another? This needs to be clarified.

Results

·         I do not see any health and economic results reported for climate change. Could this be added?

·         It seems that the climate change results are missing from Table 6, which I think is related to my earlier point about the risk perception dimensions of COVID-19 being a predictor of the risk perception dimensions of climate change. As a result, I am not confident about the findings presented.

Discussion

·         Several of the connections between risk perception related to COVID-19 and climate change raised here should be included in the introduction to strengthen the rationale for the study.

·         As with the results, it is not clear why there is only a focus on the health and economic consequences of the pandemic.

It is difficult to fully assess the discussion of the findings as I am not confident about the findings presented.

Round 2

Reviewer 1 Report

The changes made vastly improve the paper. The clarity around risk perceptions certainly helps set the tone for the rest of the article, where before it was difficult to detect a thread throughout the work.

There are still, however, some issues with the paper. I'm still struggling to see why the paper is titled as it is, but has the association between climate change and the pandemic as its secondary aim and focus in the paper. This seems a little confused at times. Unfortunately I don't feel the authors' response, that the 'chosen order of the aims follows the structure of the study' is sufficient to alleviate this. I invite further comment on this - it feels a bit like the association is being forced into a paper that is ostensibly about the pandemic, first and foremost. Could a slight reframing of the issue help here? So not placing them on equal footing in the title, but making it a piece about the pandemic with links to economic, health and other concerns (including those of another crisis)? I realise this would take substantial revision to the introduction if so.

The implications for risk communication section still talks predominantly about other research. It does seem to build up a discussion of knowledge, in order to then discuss the important role of emotion and link to this study, but this takes a long time - could this be shortened to lead into the implications of your research? Or perhaps the knowledge side could be built beforehand - indeed, some of that literature might be better placed in the previous section (i.e. 4.2). 

As above, the flow of the piece has significantly improved. However, there are still places where this could be further improved. 

Other comments: 

- The point made at 86-87 about COVID as a result of environmental degradation needs to be evidenced with a reference.

- You appear to lose a fair bit of your sample in the regression, how does this impact on the representativeness of your sample? Does it cause you to deviate from the quotas?

- I think the conclusion could do more to reflect the main discussion points of the paper.

- 417 onwards: unsure what 'mitigation' refers to in this context - could be clearer

- Are there other ways to interpret a negative framing, other than it being helpful? For example, there is a lot of literature around the pitfalls of alarmism and the de-sensitisation of repetitive negative frames.

Reviewer 2 Report

I may have missed this, but I would have expected a point-by-point response to the review comments. I still have strong concerns about the paper, as some points remain unaddressed.

Main concerns:

* It is still not clear about how the individuals were selected. I understand that there is a quota that was used to select the sample size. How was this quota determined? Why 1000? How were the participants selected based on socioeconomic variables? Were they randomly selected within socioeconomic strata? How was this considered in the analysis? Further details about the selection of the participants is needed.

* More detail about the CFA is needed. Do these represent unstandardized loadings? Is this why some loadings were greater than 1? Could some of the other fit statistics (beyond Cronbach’s alpha) be added here

* It is still not clear how the multivariate multiple linear regression analysis was conducted as no additional details were provided. Table 6 is particularly confusing as I would have expected coefficients to be reported for both climate change and COVID risk perception.